# A Review of Insights on Vaccination Against Respiratory Viral Infections in Africa: Challenges, Efforts, Impacts, and Opportunities for the Future

**DOI:** 10.3390/vaccines13090888

**Published:** 2025-08-22

**Authors:** Paul Gasana, Noel Gahamanyi, Augustin Nzitakera, Frédéric Farnir, Daniel Desmecht, Leon Mutesa

**Affiliations:** 1Department of Biomedical Laboratory Sciences, School of Health Sciences, College of Medicine and Health Sciences, University of Rwanda, Remera Campus-KG 11 Ave. 47, Kigali P.O. Box 4285, Rwanda; p.gasanal.mutesa@ur.ac.rw (P.G.); a.nzitakera@ur.ac.rw (A.N.); 2Department of Genetics, Faculty of Veterinary Medicine, University of Liège, 4000 Liège, Belgium; f.farnir@uliege.be; 3National Reference Laboratory, Rwanda Biomedical Centre, Kigali P.O. Box 7162, Rwanda; noel.gahamanyi@rbc.gov.rw; 4Unit of Pathology, Department of Morphology and Pathology, Faculty of Veterinary Medicine, University of Liège, 20 Boulevard de Colonster, 4000 Liège, Belgium; daniel.desmecht@uliege.be; 5Center for Human Genetics, School of Medicine and Pharmacy, College of Medicine and Health Sciences, University of Rwanda, Remera Campus-KG 11 Ave. 47, Kigali P.O. Box 4285, Rwanda

**Keywords:** respiratory viral infections, vaccine access, Africa, national immunization programs, vaccine hesitancy, local vaccine production

## Abstract

**Background**: Respiratory viral infections such as influenza, COVID-19, and respiratory syncytial virus (RSV) are considered as major public health threats in Africa. Despite global advancements in vaccine development, persistent inequities in access, delivery infrastructure, and public trust limit the continent’s capacity to control these diseases effectively. This review aimed at providing insights on challenges, efforts, impacts, and opportunities for the future related to vaccination against respiratory viral infections in Africa. **Methods**: This narrative review synthesizes the peer-reviewed literature and global health reports to examine vaccination efforts against respiratory viruses in Africa. The analysis focuses on disease burden, vaccine coverage, barriers to uptake, enabling factors, progress in local vaccine production, and strategies for integrating vaccines into national immunization programs (NIPs). **Results**: Respiratory vaccines have significantly reduced hospitalizations and mortality among high-risk groups in African countries. Nonetheless, key challenges, including limited cold chain capacity, vaccine hesitancy, donor-reliant supply chains, and under-resourced health systems, continue to undermine vaccine delivery. Successful interventions include community mobilization, use of mobile health technologies, and leveraging existing immunization platforms. Emerging initiatives in local vaccine manufacturing, including Rwanda’s modular mRNA facility and Senegal’s Institut Pasteur, signal a shift toward regional self-reliance. **Conclusions**: Maximizing the impact of respiratory vaccines in Africa requires a multifaceted strategy: integrating vaccines into NIPs, strengthening domestic production, expanding cold chain and digital infrastructure, and addressing sociocultural barriers through community-driven communication. These efforts are essential to achieving vaccine equity, health resilience, and pandemic preparedness across the continent.

## 1. Introduction

Globally, respiratory viral infections are among the predominant viruses associated with increased morbidity and mortality [1]. The most prevalent viruses affecting the respiratory system are the influenza virus, the respiratory syncytial virus (RSV), the parainfluenza virus, the adenovirus, the human metapneumovirus, and the enterovirus [2]. The infection is mostly limited to the upper airway and is self-limiting, but sometimes it can progress to lower respiratory tract infections such as bronchiolitis and pneumonia [3]. Each year, about 1 billion people contract seasonal flu, with 3 to 5 million experiencing severe symptoms. This leads to an estimated 290,000 to 650,000 deaths globally due to respiratory complications [4]. These viruses disproportionately affect vulnerable populations, including children, the elderly, and immunocompromised individuals, particularly in low- and middle-income countries (LMICs) [5,6]. Among the regions with the highest burden of respiratory infections, Africa stands out due to its fragile healthcare systems, limited disease surveillance infrastructure, and historical inequities in vaccine access [7,8].

Vaccination remains one of the most effective public health interventions to prevent morbidity and mortality associated with respiratory viral diseases [9]. Vaccination is also known to improve life expectancy and economic growth [10]. The World Health Organization (WHO) estimates that immunization prevents 4 to 5 million deaths annually [11], of which 2–3 million are children [12,13]. For respiratory infections, vaccines work by priming the immune system through inactivated, live-attenuated, subunit, or genetic platforms (e.g., mRNA), promoting protective immune responses [14,15].

Influenza vaccines have been available for over 50 years and are crucial in lessening the impact of seasonal epidemics [15]. However, their high cost and the need for annual revaccination limit their use in low- and middle-income countries (LMICs) [2]. While significant progress has been made with RSV vaccines currently under clinical evaluation [16], early inactivated RSV vaccines led to severe disease in vaccinated individuals, complicating a swift solution [17]. Unfortunately, no vaccines are yet available for other respiratory viruses [2].

The emergence of coronavirus disease 2019 (COVID-19) accelerated the global development of various vaccine platforms, including mRNA (e.g., Pfizer-BioNTech, Moderna), viral vectors (e.g., Oxford-AstraZeneca, Johnson & Johnson), protein subunits (e.g., Novavax), and inactivated virus vaccines (e.g., Sinopharm, Sinovac) [18,19,20,21]. Despite remarkable scientific advancements in vaccine development, African countries face considerable challenges in implementing effective vaccination programs. These efforts in Africa are hindered by socioeconomic challenges such as poverty, poor sanitation, and overcrowding, as well as weak health systems with limited surveillance, inadequate infrastructure, and logistical barriers [22]. Moreover, the COVID-19 pandemic exposed global inequities in vaccine distribution and revealed the urgent need for regional vaccine manufacturing and self-reliance [23,24].

This review aims to synthesize current evidence on vaccination efforts against respiratory viral infections across the African continent. Specifically, it examines the effectiveness of vaccines, implementation strategies, barriers to access, regional manufacturing initiatives, and the public health impact of vaccination programs. It also explores current challenges and potential opportunities to inform future vaccine strategies and improve pandemic preparedness and health equity in Africa.

## 2. Materials and Methods

A comprehensive literature search was systematically conducted across major academic databases, including PubMed, Scopus, and Google Scholar. The search aimed to identify relevant studies on vaccination efforts against respiratory viral infections in Africa. The following keywords and their combinations were used: “vaccination Africa” OR “immunization Africa” AND (“influenza vaccine” OR “RSV vaccine” OR “respiratory syncytial virus vaccine” OR “COVID-19 vaccine” OR “SARS-CoV-2 vaccine”) AND (“effectiveness” OR “coverage” OR “uptake” OR “barriers” OR “facilitators” OR “local production” OR “policy”).

The inclusion criteria for articles were as follows: original research articles, policy documents, and reports published in English focusing on vaccination efforts against influenza, RSV, or SARS-CoV-2 in African countries. The exclusion criteria included (i) opinion pieces, editorials, and conference abstracts without full papers and (ii) studies not specifying the respiratory viruses assessed or the period during which the study was conducted.

Articles were initially selected based on a thorough review of their titles and abstracts. To minimize bias, the literature was independently reviewed by two authors. In cases of disagreement regarding article inclusion, a third author was consulted as a tie-breaker to reach a consensus.

The discussion was constructed based on a thematic synthesis of the selected literature, focusing on the following themes: vaccine effectiveness in African populations, vaccine coverage and immunization strategies, barriers and facilitators to vaccine uptake, the impact of vaccination programs on public health outcomes, and progress in local vaccine production and policy initiatives.

## 3. Results

A comprehensive literature search yielded 2723 citations from multiple databases and relevant sources. After initial screening of titles and abstracts, 238 articles were deemed potentially relevant to the topic of vaccination against respiratory viral infections in Africa. Following this stage, 57 articles were excluded due to their lack of relevance to vaccination issues within the African context.

Full texts of the remaining 181 articles were examined in greater detail. During this process, 129 articles were excluded based on the following criteria: 42 were duplicates, 18 fell outside the study’s predefined publication timeframe, 57 were unrelated to respiratory viral vaccines in Africa, 1 was an editorial, 1 an opinion piece, 1 a conference abstract, and 9 were excluded for poor methodological rigor or insufficient data quality.

An additional 16 articles were excluded after further review for not adequately addressing the key dimensions of the review, namely, the challenges, efforts, impacts, or future opportunities related to vaccination against respiratory viral infections in Africa. Moreover, six articles were found to be redundant or overlapping in content, and two full texts were inaccessible despite multiple retrieval attempts. Ultimately, 28 studies were included in this narrative review.

These studies were geographically diverse, covering regions across Sub-Saharan Africa and North Africa, and addressed a broad range of respiratory viral infections such as influenza, respiratory syncytial virus (RSV), and SARS-CoV-2. The selected literature provided insights into vaccination strategies, implementation challenges, epidemiological patterns, vaccine coverage, public health policies, and barriers to equitable vaccine access across the continent (Table 1).

## 4. Discussion

Vaccination remains a cornerstone of public health, especially in the context of respiratory viral infections, which continue to impose a significant burden across Africa. This discussion synthesizes current evidence on vaccine effectiveness, coverage, implementation strategies, and barriers within the African context, drawing attention to both the progress made and persistent challenges (Table 2) [28].

### 4.1. Vaccine Effectiveness in African Populations

Vaccines targeting respiratory viruses have shown substantial benefits in African populations, contributing to reductions in morbidity and mortality, particularly among high-risk groups. However, their effectiveness varies due to differences in circulating viral strains, host immune responses, comorbidities, and challenges in vaccine uptake and distribution across the continent [7,24].

Influenza vaccination has demonstrated moderate but meaningful effectiveness in reducing the burden of influenza-like illness (ILI), particularly among healthcare workers, pregnant women, the elderly, and individuals with comorbidities. For instance, a multi-year study in South Africa estimated seasonal influenza vaccine effectiveness (VE) ranging from 30% to 50%, aligning with global averages [31,32]. Another study in Kenya among pregnant women reported a significant reduction in laboratory-confirmed influenza cases among both mothers and their infants following maternal immunization [33].

One particularly compelling study that illustrates the strategic value of maternal immunization was conducted in Kenya by Nunes et al. [33]. This randomized, double-blind, placebo-controlled trial enrolled over 2000 pregnant women to evaluate the effectiveness of maternal influenza vaccination in preventing laboratory-confirmed influenza in both mothers and their infants. The study found that maternal vaccination conferred 50% efficacy in mothers and 61% efficacy in their infants during the first six months of life—a period in which infants are at high risk for severe influenza complications yet are ineligible for direct vaccination. The dual protection demonstrated by this study not only underscores the biological effectiveness of maternal vaccination but also its public health significance in resource-limited African settings, where infant morbidity and mortality due to respiratory viruses remain high and access to postnatal healthcare may be limited. This case exemplifies how targeted vaccine strategies adapted to population vulnerabilities can achieve substantial impact even where broader vaccine infrastructure is still developing.

Despite these benefits, influenza vaccine uptake in Africa remains low due to high costs, limited integration into national immunization programs, and insufficient public awareness [34].

The COVID-19 vaccines have provided robust protection against severe disease, hospitalization, and death in African populations, despite logistical and distribution challenges. Real-world effectiveness studies during the Delta and Omicron waves demonstrated that mRNA vaccines (Pfizer-BioNTech and Moderna) and viral vector vaccines (Oxford-AstraZeneca and Johnson & Johnson) retained substantial effectiveness in preventing severe COVID-19 outcomes [35]. In South Africa, data from the Sisonke study, which administered the Johnson & Johnson vaccine to nearly 500,000 healthcare workers under real-world programmatic conditions, showed 67% effectiveness against hospitalization and over 90% effectiveness against death [36]. Even during variant-driven surges, vaccinated individuals consistently experienced better outcomes, underscoring the importance of expanding vaccine coverage across the continent.

The Sisonke study is particularly compelling as it illustrates the feasibility and impact of large-scale vaccine deployment in high-risk populations during a public health emergency. Conducted during waves dominated by the Beta and Delta variants, the study demonstrated not only the efficacy of a single-dose viral vector vaccine but also the potential for rapid, coordinated vaccine rollout in resource-constrained settings. The program leveraged South Africa’s existing healthcare infrastructure and targeted a frontline population, generating critical real-world evidence and reinforcing the importance of locally led implementation efforts during global pandemics.

Even during variant-driven surges, vaccinated individuals consistently experienced better outcomes, underscoring the importance of expanding vaccine coverage across the continent.

For respiratory syncytial virus (RSV), evidence on vaccine effectiveness in African populations remains limited, primarily due to the historical absence of licensed vaccines. However, clinical trials in Kenya and South Africa have evaluated the safety and immunogenicity of RSV vaccine candidates, including maternal vaccines and monoclonal antibodies (e.g., nirsevimab), with encouraging results [37,38]. A recent multicounty trial demonstrated that maternal RSV vaccination significantly reduced medically attended RSV-associated lower respiratory tract infections in infants, including participants from South Africa [39]. These findings highlight the potential of RSV vaccines to substantially reduce infant morbidity and mortality, particularly in high-burden African settings.

### 4.2. Vaccine Coverage and Immunization Strategies

Vaccine coverage across African nations remains markedly uneven, reflecting a complex interplay of socioeconomic, political, and infrastructural factors [8]. While influenza vaccination is formally recommended in a few African countries for high-risk populations, such as healthcare workers, the elderly, pregnant women, and individuals with chronic illnesses, it is not yet integrated into most national immunization programs (NIPs). This omission stems from budgetary constraints, inadequate vaccine supply chains, limited local production capacity, and competing healthcare priorities such as malaria, HIV/AIDS, and tuberculosis [40].

The COVID-19 pandemic underscored various structural weaknesses. During the initial phase of vaccine deployment, most African countries heavily relied on multilateral initiatives, such as COVAX, as well as bilateral donations due to their limited purchasing power and local manufacturing capabilities [41]. Although these mechanisms facilitated access to vaccines, actual coverage rates remained disappointingly low. By early 2023, only about 25% of Africa’s population had received at least one dose of a COVID-19 vaccine, substantially below the global average of over 70% in high-income regions [42,43]. The disparity highlights both supply-side issues and deep-rooted demand-side challenges, including vaccine hesitancy, misinformation, and distrust in public health institutions [44,45].

Immunization strategies in Africa typically involve a combination of periodic mass vaccination campaigns and routine immunization services delivered through primary healthcare networks. Community health workers play a pivotal role in vaccine outreach, especially in remote and underserved areas [46]. However, the effectiveness of these strategies is undermined by fragile health information systems, irregular funding, insufficient human resources, and inadequate cold chain infrastructure necessary to store and transport temperature-sensitive vaccines [47,48]. For example, in rural settings where the electricity supply is unreliable, maintaining proper vaccine storage conditions becomes exceedingly difficult, risking vaccine spoilage and loss [49].

Innovative approaches, such as the use of solar-powered refrigerators, digital health tools for real-time data tracking, and mobile clinics, are being piloted in several countries with promising results [50]. Moreover, integrating immunization services with maternal and child health programs has proven to increase vaccine uptake in certain regions. Still, scaling these interventions requires sustained investment, political will, and robust international partnerships.

Strengthening vaccine coverage and immunization strategies in Africa calls for a holistic approach—one that not only improves supply and distribution infrastructure but also addresses sociocultural barriers and fosters public trust in vaccines. Targeted policy reforms, regional collaboration, and increased local vaccine production are essential to achieving equitable and resilient immunization systems across the continent [24].

### 4.3. Barriers and Facilitators to Respiratory Disease Vaccination in Africa

A complex interplay of logistical, sociocultural, and systemic factors influences vaccine uptake across Africa. Persistent structural and infrastructural limitations pose significant barriers, particularly for vaccines that require stringent storage conditions, such as mRNA vaccines that necessitate ultra-cold chain equipment. This often leads to delivery delays, stockouts, and limited access in rural and underserved areas due to insufficient transportation and the uneven distribution of vaccination centers [51,52,53].

In addition, Africa’s limited domestic manufacturing capacity creates a reliance on multilateral mechanisms like COVAX and bilateral donations, which can delay access to adequate vaccine supplies and expose vulnerabilities in self-sufficiency [6,54]. These issues are compounded by inconsistent data systems and under-resourced national immunization programs, hindering equitable distribution and monitoring. Beyond these structural challenges, behavioral and sociocultural elements significantly impact vaccine demand. Vaccine hesitancy, fueled by misinformation, low risk perception, and mistrust in public institutions, remains a significant hurdle. Myths about fertility and vaccine safety, often spread via social media, negatively impact public attitudes [53]. Historical experiences with unethical medical research and perceptions of foreign dominance in vaccine development further exacerbate public skepticism in many communities [54,55].

Despite these multifaceted challenges, there are encouraging signs of progress, with several facilitating factors proving effective in enhancing vaccine uptake. Community engagement strategies, particularly involving religious and traditional leaders, have been instrumental in building trust and countering misinformation [56]. Health education campaigns tailored to local languages and cultural contexts have also increased vaccine acceptance in diverse settings [57]. Countries such as Rwanda, Ghana, and Senegal have successfully leveraged existing robust immunization infrastructure, including mobile health technologies and SMS-based reminders, to improve outreach and uptake [40,56]. The strategic use of community health workers and mobile clinics extended services to marginalized populations, effectively bridging both supply and demand gaps [58].

Achieving sustainable improvements in vaccine uptake across Africa requires a multifaceted and context-sensitive approach. Strengthening health systems, expanding domestic vaccine production, enhancing data systems, and fostering local ownership of immunization campaigns are essential to overcoming persistent barriers. Regional collaboration, transparent communication, and long-term investment will be key to realizing equitable and resilient vaccination programs across the continent.

### 4.4. Impact of Vaccination Programs on Public Health Outcomes

Vaccination programs targeting respiratory viruses have played a pivotal role in reducing disease burden, mortality, and the strain on healthcare systems across African nations [59]. These interventions have demonstrated notable success, particularly in protecting high-risk groups such as healthcare workers, the elderly, pregnant women, and individuals with comorbidities [60].

Influenza vaccination has been shown to lower the incidence of severe influenza-like illness, reduce hospitalizations, and prevent mortality during seasonal peaks. In countries like South Africa and Kenya, influenza vaccination among priority populations led to significant declines in laboratory-confirmed influenza and associated complications [61]. Another study estimated that influenza vaccines reduced the risk of hospitalization by 30–60% in older adults when coverage was adequate [62].

The COVID-19 vaccination has further highlighted the benefits of widespread immunization. Despite initial inequities in vaccine access, studies from countries such as South Africa, Morocco, and Rwanda have reported substantial reductions in severe COVID-19 outcomes [63,64]. Real-world data from the Sisonke study in South Africa indicated that the Johnson & Johnson vaccine reduced COVID-19-related hospitalization by 67% and mortality by over 90% among healthcare workers [36]. Moreover, vaccination efforts contributed to mitigating the impact of successive waves of variants such as Delta and Omicron by preventing severe cases and reducing ICU admissions [65].

Beyond direct health outcomes, these vaccination programs have significantly reduced the economic and systemic burden associated with respiratory outbreaks. By decreasing disease transmission, vaccines have helped maintain healthcare system functionality and reduced productivity losses and treatment costs, particularly critical in resource-constrained African settings [10,65].

In the case of respiratory syncytial virus (RSV), the absence of a licensed vaccine has limited disease control options. Nevertheless, public health interventions—such as promoting hand hygiene, respiratory etiquette, use of personal protective equipment (PPE), and health education—remain essential in mitigating RSV transmission [10]. Promising developments are underway, including maternal RSV vaccine trials and monoclonal antibodies (e.g., nirsevimab), with recent Phase III trials conducted in African countries showing encouraging efficacy and safety profiles [39]. These findings collectively affirm the importance of sustained investment in vaccination infrastructure and innovation. Tailoring immunization programs to local epidemiological profiles and ensuring equitable access are key to maximizing the long-term health and economic benefits of vaccines in Africa.

### 4.5. Progress in Local Vaccine Production and Policy Initiatives

Building sustainable vaccine production in Africa requires significant investment in several key areas. First, infrastructure development is crucial, encompassing state-of-the-art manufacturing facilities, quality control labs, cold chain storage, and reliable utilities like electricity and clean water. Rwanda in partnership with BioNTech is building a modular mRNA vaccine production facility aimed at boosting regional vaccine access with COVID-19 as the starting point [66]. Second, technology transfer and intellectual property are vital, necessitating investments in licensing agreements and supporting initiatives like the WHO’s mRNA vaccine technology transfer hub [67]. South Africa’s Biovac and Aspen Pharmacare have partnered in vaccine fill-finish and manufacturing, contributing to COVID-19 vaccine availability [68]. Senegal’s Institut Pasteur de Dakar is advancing toward full-scale vaccine production, including mRNA-based technologies [69]. Furthermore, the Biovac institute through Afrigen Biologics will transfer the COVID-19 mRNA vaccine to companies in Senegal (IPD), Nigeria (Biovaccines Nigeria), Tunisia (IPT), and Egypt (BioGeneric Pharma) [70]. Alongside this, funding for local Research and Development (R&D) is essential to adapt existing technologies and develop new vaccines tailored to regional needs, along with conducting clinical trials within Africa [24]. Third, human capital development is paramount, and substantial investment is needed in specialized training for scientists, engineers, and technicians, along with retention strategies [71]. Fourth, regulatory harmonization and strengthening are critical. Funding and technical support for national regulatory authorities and initiatives like the African Medicines Agency (AMA) will ensure quality and streamline distribution. Finally, governments must foster Public–Private Partnerships (PPPs), establish dedicated funds, and provide long-term procurement guarantees to create market certainty. Policy frameworks such as the African Union’s Partnerships for African Vaccine Manufacturing (PAVM) and WHO’s mRNA vaccine technology transfer hub initiative represent pivotal efforts to support sustainable immunization systems on the continent [24].

### 4.6. Actionable Steps for Integrating Respiratory Viral Vaccines into National Immunization Programs (NIPs)

Building sustainable local vaccine production capabilities in Africa requires significant and sustained investment across multiple key domains. Fundamental to this endeavor is infrastructure development, which encompasses the construction and equipping of state-of-the-art manufacturing facilities [72]. These facilities must include sterile manufacturing suites, advanced quality control laboratories, and specialized cold chain storage, all of which require substantial upfront capital investment [72]. Additionally, reliable utilities infrastructure such as consistent electricity, clean water supply, and efficient waste management systems is essential for effective pharmaceutical manufacturing.

Integrating respiratory viral vaccines, such as those for influenza, RSV, and COVID-19, into existing National Immunization Programs (NIPs) in Africa requires a strategic, evidence-based approach [52]. The foundational step involves conducting comprehensive studies of disease burden to prioritize the most challenging ones due to limited resources [53]. For example, influenza continues to cause significant seasonal morbidity and mortality, particularly in children under five and the elderly. RSV is a leading cause of lower respiratory tract infections in infants and young children, while COVID-19 has disproportionately affected older adults and individuals with comorbidities [73,74].

In terms of age-specific vaccine administration, COVID-19 vaccines are typically recommended for individuals aged 12 years and above, with many countries in Africa expanding eligibility to children aged 5–11 years. Influenza vaccination is advised annually for high-risk groups, including children aged 6 months to 5 years, pregnant women, and the elderly. The RSV vaccine landscape has evolved with the approval of nirsevimab—a long-acting monoclonal antibody—for neonates and infants, including preterm babies. Additionally, maternal RSV vaccination (during the third trimester) to confer passive immunity to newborns is gaining approval in several countries [75,76].

Currently, these vaccines are not universally included in all African countries’ National Immunization Schedules. Influenza vaccines are typically administered through targeted campaigns rather than routine immunization schedules. COVID-19 vaccines were rapidly integrated under emergency use protocols but are now facing transition challenges in routine uptake. RSV vaccines, particularly for infants and pregnant women, are only beginning to be considered for inclusion, with pilot programs and donor-supported initiatives underway in selected countries [77].

In addition to viral vaccines, integrating bacterial respiratory vaccines such as pneumococcal conjugate vaccines (PCVs) and hexavalent vaccines into NIPs has also yielded critical public health benefits in Africa. PCVs have significantly reduced the burden of pneumonia, bacteremia, and meningitis caused by Streptococcus pneumoniae, particularly in children under five [78,79]. The hexavalent vaccine, which protects against six diseases (diphtheria, tetanus, pertussis, hepatitis B, Haemophilus influenzae type b, and poliovirus), streamlines the immunization schedule and improves adherence while addressing multiple pathogens contributing to respiratory morbidity [80,81]. The incorporation of these vaccines into national schedules, as seen in countries like Rwanda, Kenya, and South Africa, offers valuable lessons for the integration of viral respiratory vaccines by leveraging existing delivery platforms, community trust, and healthcare infrastructure [70,82,83].

Public acceptance varies significantly across the continent. For example, Rwanda’s COVID-19 vaccination campaign achieved one of the highest coverage rates in Africa through strong political leadership, the deployment of trusted community health workers, mobile vaccination units, and localized public messaging strategies [84]. Similarly, Senegal’s polio campaigns overcame community resistance through mosque-based communication, door-to-door outreach, and engagement of religious leaders, which can serve as a model for future respiratory vaccine rollouts [85].

Securing sustainable funding mechanisms and strong supply chain management are paramount for successful integration. This involves actively integrating vaccine procurement and delivery costs into national health budgets, which may necessitate increasing domestic health expenditure [54]. Countries should also pursue favorable pricing negotiations with vaccine manufacturers and explore innovative financing mechanisms, including partnerships with global health initiatives [86]. A robust supply chain and cold chain infrastructure are indispensable. This requires significant investment in reliable cold chain equipment, including specialized ultra-cold storage for certain mRNA vaccines, coupled with ensuring a consistent electricity supply. The supply chain system can be strengthened by using robust health information systems, including digital health platforms for monitoring vaccine coverage in real-time, tracking adverse events, and identifying underserved populations to adapt strategies dynamically. Improving transportation networks is essential to guarantee that vaccines can reach all communities, including remote and hard-to-reach areas, complemented by efficient inventory management systems to prevent stockouts and reduce waste.

Lastly, enhancing the health workforce is essential to improving vaccination outcomes, particularly in regions like Africa, where there is a critical shortage of skilled professionals [70]. This involves not only introducing postgraduate diploma and certificate programs, as well as one- to two-year master’s degree courses in vaccinology, but also providing comprehensive training in key areas such as vaccine administration, cold chain management, monitoring of adverse events, and effective communication to address vaccine hesitancy. Together, these efforts can significantly strengthen workforce capacity and support more effective immunization programs. Community health workers are particularly vital in outreach efforts, building trust and promoting vaccine acceptance. Targeted communication and community engagement strategies are also essential, focusing on culturally sensitive health education campaigns to counter misinformation and involving trusted local leaders [56].

An example of a successful program addressing vaccine hesitancy comes from Rwanda’s COVID-19 vaccination campaign, which achieved high coverage about 77.9% fully vaccinated by September 2022, through a trusted network of community health workers, tailored messaging, and strong political leadership. Rwanda’s use of community-based delivery, trust in health professionals, and the behavioral 3Cs framework (confidence, complacency, convenience) helped overcome hesitancy and build public trust [87,88].

Finally, a phased rollout, beginning with pilot programs in specific regions or for high-risk groups, allows for the gathering of data, identification of operational challenges, and refinement of strategies before a broader, nationwide implementation [58]. Ensuring national regulatory authorities are equipped to efficiently review and approve new vaccines, aligning with international standards, underpins the entire integration process.

A key limitation noted in this review is the relatively limited availability of Africa-specific research on vaccines for respiratory infections such as influenza, COVID-19, and RSV. Many studies are concentrated in a few countries or conducted in non-African settings, which may not fully capture the continent’s regional diversity of challenges, implementation strategies, and outcomes. This highlights the ongoing need for more locally generated evidence to inform vaccine policy and practice tailored to Africa’s unique contexts.

## 5. Conclusions

Africa continues to experience waves of respiratory diseases like influenza, COVID-19, and RSV. Several challenges in vaccine access, storage, distribution, acceptance, limited financial and healthcare resources have been highlighted. Efforts to fill in the gaps include the establishment of vaccine manufacturing facilities on African continent through technology transfer and collaboration with international organizations supporting increased vaccine coverage. Also, increase in training programs tailored to identified challenges and strengthening regulatory frameworks could boost the vaccination coverage and reduce the burden due to vaccine-preventable diseases. Expanding vaccine education campaigns and embedding trust-building into health policy are also critical for improving long-term vaccine uptake. We recommend integrating respiratory viral vaccines into NIPs and adopting digital platforms in monitoring vaccine coverage and use for better pandemic preparedness across Africa.

## Figures and Tables

**Table 1 vaccines-13-00888-t001:** Key insights on vaccination against respiratory virus diseases in Africa.

	Subtopic	Description	Gaps	References
1.	Vaccine Effectiveness in African Populations	The effectiveness of influenza and COVID-19 vaccines varies by type and population. Moderate to high efficacy observed in trials.	Limited local clinical trials; diversity in genetic and nutritional factors not fully studied.	[25]
2.	Vaccine Coverage and Immunization Strategies	Coverage is higher in urban regions. Integration with EPI has improved uptake in some countries.	Rural areas have limited access and inadequate data on strategy effectiveness.	[8]
3.	Barriers and Facilitators to Respiratory Disease Vaccination in Africa	Barriers include misinformation, hesitancy, and supply issues. Facilitators include government support and international aid.	Persistent vaccine hesitancy; insufficient investment in health communication.	[7]
4.	Impact of Vaccination Programs on Public Health Outcomes	Vaccination reduces disease burden, hospitalizations, and supports pandemic response.	Lack of consistent monitoring and evaluation systems.	[25]
5.	Progress in Local Vaccine Production and Policy Initiatives	Efforts are underway in countries like Senegal and South Africa to produce vaccines locally.	Insufficient capacity and regulatory frameworks; reliance on imports remains high.	[24]
6.	Actionable Steps for Integrating Respiratory Viral Vaccines into NIPs	Requires stakeholder engagement, resource alignment, and strategic planning.	Need for structured national plans and sustainable funding.	[26,27]

**Table 2 vaccines-13-00888-t002:** Dissemination strategies, implementation challenges, and funding implications for respiratory viral vaccines in African contexts.

No.	Virus and Vaccine Status	Dissemination Efforts in Africa	Key Challenges and Barriers	Impact of Foreign Aid and Policy	References
1.	Influenza—licensed and updated seasonally	Seasonal vaccination campaigns in select countries; limited integration into national immunization programs (NIPs)	Limited local production; low awareness; irregular vaccine availability; antigenic drift reduces effectiveness	Decreased international funding threatens sustained campaigns and vaccine procurement; reliance on global donors remains high	[8,29]
2.	COVID-19—widely distributed with emergency and full use authorizations	Large-scale vaccination drives supported by COVAX, Africa CDC initiatives, and national campaigns; expanding booster coverage	Cold chain logistics (especially for mRNA vaccines), vaccine hesitancy, uneven coverage between urban and rural areas, supply inconsistencies	Recent cutbacks in foreign aid and shifting global priorities risk slowing vaccine access; local manufacturing efforts are underway but still limited	[18]
3.	RSV—recently approved (e.g., Arexvy for older adults)	Pilot maternal immunization programs and clinical trials underway; limited rollout in pediatric populations	High vaccine cost; regulatory hurdles; lack of awareness; insufficient healthcare infrastructure for wide-scale deployment	Global support is crucial for introduction; reductions in aid may delay access; partnerships with GAVI and WHO are critical for future scaling	[30]

## Data Availability

Not applicable.

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
