# Peer review of "A Review of Insights on Vaccination Against Respiratory Viral Infections in Africa: Challenges, Efforts, Impacts, and Opportunities for the Future"

_vaccines, 2025, doi:10.3390/vaccines13090888_

Round 1
Reviewer 1 Report
Comments and Suggestions for Authors
Considering the magnitude of the problem and the size of the continent, this review is relatively concise and still brings up the major issues affecting vaccination in Africans. By necessity given its limited scope, it is mostly superficial. While there is nothing wrong with being brief, a few more specific examples, perhaps for each of the major topics (maybe as a box or a figure) would make this manuscript more engaging. Additionally, the tables can be designed to be more appealing and readable. I had trouble figuring out which lines fit which comment.
- The title should reflect that this is a review of the literature; I am not sure what they mean by insights.
- In keeping with that, again, describe in more detail a particular paper that the authors found compelling in illustrating a specific principle. This will add a lot of value to the manuscript.
- The paragraph beginning in line 81 of the manuscript should be a summary of the goals. The readers can decide whether the review contributes to the field (line 81).
- Obviously, line 375 is a typo. No need to tell the reader that the section that follows is redundant, inviting them not to read it.
- Table 2 is not particularly informative. Readers of this journal know which vaccines are available in general (and is there anyone on the planet who does not know there is a COVID vaccine?). Maybe some expansion of the current efforts to disseminate them in Africa would be more interesting. At the risk of being political, maybe even a comment about how current cutbacks in foreign aid will affect distribution.
- Perhaps an example of a successful program to overcome vaccine hesitancy would also be useful.
Author Response
Comments and Suggestions for Authors
Comment 1: Considering the magnitude of the problem and the size of the continent, this review is relatively concise and still brings up the major issues affecting vaccination in Africans. By necessity given its limited scope, it is mostly superficial. While there is nothing wrong with being brief, a few more specific examples, perhaps for each of the major topics (maybe as a box or a figure) would make this manuscript more engaging. Additionally, the tables can be designed to be more appealing and readable. I had trouble figuring out which lines fit which comment.
Answer 1: We thank the reviewer for appreciating our manuscript. We made considerable changes to the revised manuscript.
Comment 2: The title should reflect that this is a review of the literature; I am not sure what they mean by insights (…).
Answer 2: Thank you for your valuable feedback. We revised the title to of the manuscript to read: Vaccination Against Respiratory Viral Infections in Africa: Challenges, Efforts, Impacts, and Future Opportunities: A Review of the Literature
Comment 3: In keeping with that, again, describe in more detail a particular paper that the authors found compelling in illustrating a specific principle. This will add a lot of value to the manuscript.
Answer 3: We thank the reviewer for the insightful suggestion. We expanded Section 4.1 to include two detailed examples of compelling studies that effectively illustrate key principles discussed in the review.
Comment 4: The paragraph beginning in line 81 of the manuscript should be a summary of the goals. The readers can decide whether the review contributes to the field (line 81).
Answer 4: We thank the reviewer for the input and we revised the paragraph.
Comment 5: Obviously, line 375 is a typo. No need to tell the reader that the section that follows is redundant, inviting them not to read it.
Answer 5: Thank you for pointing out the issue regarding the content at line 375. We have revised Section 4.6 to remove any statements that might discourage the reader from continuing, ensuring the text remains focused and engaging. The updated Section 4.6 can be found on line 350–432 of the revised manuscript.
Comment 6: Table 2 is not particularly informative. Readers of this journal know which vaccines are available in general (and is there anyone on the planet who does not know there is a COVID vaccine?). Maybe some expansion of the current efforts to disseminate them in Africa would be more interesting. At the risk of being political, maybe even a comment about how current cutbacks in foreign aid will affect distribution.
Answer 6: We appreciate the reviewer’s insightful feedback regarding Table 2. We revised Table 2 to focus on the current challenges, dissemination efforts, and context-specific issues affecting vaccine deployment in Africa. The revised Table 2 and the expanded discussion can be found in Section 4.6, line 337 of the updated manuscript.
Comment 7: Perhaps an example of a successful program to overcome vaccine hesitancy would also be useful.
Answer 7: We thank the reviewer for the feedback. We revised Section 4.6 to include real-world examples from Rwanda’s COVID-19 vaccination campaign.
Reviewer 2 Report
Comments and Suggestions for Authors
It would be extremely interesting if the authors could comment also on the age when different vaccines are received. It would be interested also to discuss the existence in the National Immunization Schedule of the vaccines discussed in the article. What is the acceptaance in the general population of any of the vaccines?It would also be interesting to discuss in more depth the incidence and prevalence of the respiratory diseases influenza, RSV etc.
Could the authors comment on the acceptance of RSV vaccine for premature babies the new Nirsevimab and also the maternal vaccination.
Author Response
Comments and Suggestions for Authors
Comment 1: It would be extremely interesting if the authors could comment also on the age when different vaccines are received. It would be interested also to discuss the existence in the National Immunization Schedule of the vaccines discussed in the article. What is the acceptance in the general population of any of the vaccines? It would also be interesting to discuss in more depth the incidence and prevalence of the respiratory diseases influenza, RSV etc. Could the authors comment on the acceptance of RSV vaccine for premature babies the new Nirsevimab and also the maternal vaccination.
Answer 1: We sincerely thank the reviewer for these thoughtful and constructive suggestions. In response, we have substantially revised Section 4.6 to incorporate these important aspects lines 350-432
Round 2
Reviewer 1 Report
Comments and Suggestions for Authors
Accept in present form.
Author Response
We thank the reviewer for accepting the current version of our manuscript for its publication.
Reviewer 2 Report
Comments and Suggestions for Authors
Would like to thank to the authors for answering to all the questions regarding their work.
It is an important review of current respiratory vaccine uptake and implementation in Africa. The authors have been focusing mainly their work on viral respiratory vaccines but it would have been interesting to also discuss their implementationrelated to pneumococcal vaccines and also to the hexavalent ones.
Author Response
Comments and Suggestions for Authors
Comment 1: It is an important review of current respiratory vaccine uptake and implementation in Africa. The authors have been focusing mainly their work on viral respiratory vaccines but it would have been interesting to also discuss their implementation related to pneumococcal vaccines and also to the hexavalent ones.
Answer 1: We thank the reviewer for this valuable suggestion. We agree that pneumococcal and hexavalent vaccines are crucial components of respiratory disease prevention in Africa, we have added a new Paragraph in section 4.6 -Pneumococcal and Hexavalent Vaccines in Africa” and it reads as follows:
“In addition to viral vaccines, integrating bacterial respiratory vaccines such as pneumococcal conjugate vaccines (PCVs) and hexavalent vaccines into NIPs has also yielded critical public health benefits in Africa. PCVs have significantly reduced the burden of pneumonia, bacteremia, and meningitis caused by Streptococcus pneumoniae, particularly in children under five[75, 76]. The hexavalent vaccine, which protects against six diseases (diphtheria, tetanus, pertussis, hepatitis B, Haemophilus influenzae type b, and poliovirus), streamlines the immunization schedule and improves adherence while addressing multiple pathogens contributing to respiratory morbidity [77, 78]. The incorporation of these vaccines into national schedules, as seen in countries like Rwanda, Kenya, and South Africa, offers valuable lessons for the integration of viral respiratory vaccines by leveraging existing delivery platforms, community trust, and healthcare infrastructure [67, 79, 80].”